# Structural identification of *N*-glycan isomers using logically derived sequence tandem mass spectrometry

Chia Yen Liew [1,2,3], Chu-Chun Yen [1], Jien-Lian Chen[1], Shang-Ting Tsai[1], Sujeet Pawar[4,5,6], Chung-Yi Wu [4,5✉] & Chi-Kung Ni [1,3,6✉]

N-linked glycosylation is one of the most important protein post-translational modifications. Despite the importance of *N*-glycans, the structural determination of *N*-glycan isomers remains challenging. Here we develop a mass spectrometry method, logically derived sequence tandem mass spectrometry (LODES/MS$^n$), to determine the structures of *N*-glycan isomers that cannot be determined using conventional mass spectrometry. In LODES/MS$^n$, the sequences of successive collision-induced dissociation are derived from carbohydrate dissociation mechanisms and apply to *N*-glycans in an ion trap for structural determination. We validate LODES/MS$^n$ using synthesized *N*-glycans and subsequently applied this method to *N*-glycans extracted from soybean, ovalbumin, and IgY. Our method does not require permethylation, reduction, and labeling of *N*-glycans, or the mass spectrum databases of oligosaccharides and *N*-glycan standards. Moreover, it can be applied to all types of *N*-glycans (high-mannose, hybrid, and complex), as well as the *N*-glycans degraded from larger *N*-glycans by any enzyme or acid hydrolysis.

[1] Institute of Atomic and Molecular Sciences, Academia Sinica, Taipei, Taiwan. [2] International Graduate Program of Molecular Science and Technology, National Taiwan University, Taipei, Taiwan. [3] Molecular Science and Technology, Taiwan International Graduate Program, Academia Sinica, Taipei, Taiwan. [4] Genomics Research Center, Academia Sinica, Taipei, Taiwan. [5] Chemical Biology and Molecular Biophysics, Taiwan International Graduate Program, Academia Sinica, Taipei, Taiwan. [6] Department of Chemistry, National Tsing Hua University, Hsinchu, Taiwan. ✉email: cyiwu@gate.sinica.edu.tw; ckni@po. iams.sinica.edu.tw

N-linked glycosylation is one of the most important post-translational modifications of proteins. N-linked glycans stabilize protein structures, promote protein folding, and regulate protein functions[1,2]. N-linked protein glycosylation has been studied in detail in eukaryotes. The core structure of $N$-glycans consists of two $N$-acetyl glucosamine units and three mannose residues. Further glycosylation of the core $N$-glycan results in various structures that are classified into three types: high-mannose, complex, and hybrid $N$-glycans. Conventionally, nuclear magnetic resonance (NMR) spectroscopy is used for the structural determination of $N$-glycans[3–6]. However, NMR requires samples of a few milligrams; thus, it is unsuitable for samples available in minute amounts, such as $N$-glycans extracted from cancer cells or antibodies.

Mass spectrometry (MS) has sensitivity that is several orders of magnitude higher than that of NMR, and it is suitable for small sample amounts. MS is widely used for the structural determination of oligosaccharides[7–10]. However, conventional MS cannot distinguish $N$-glycan isomers, therefore, additional methods are needed for differentiation. One of the additional methods is the prior knowledge of the possible structures according to the principles and rules of glycan biosynthesis. For example, only one isomer of $Man_5GlcNAc_2$ high-mannose $N$-glycan, namely $5A_{1,2}$ (according to the nomenclature proposed by Reinhold et al[11].), is produced in eukaryotes based on the corresponding biosynthetic pathways. Consequently, if the ion of $Man_5GlcNAc_2$ is found in the MS of eukaryote samples, it is assumed to be the isomer $5A_{1,2}$[12,13]. By contrast, more than one isomer of other $N$-glycans, such as $Man_8GlcNAc_2$, $Man_7GlcNAc_2$, and asymmetrical multi-antennary complex $N$-glycans, can be generated in eukaryotes. Therefore, in addition to the method based on the principles and rules of glycan biosynthesis, a complementary method (e.g., enzyme digestion) is required to differentiate these isomers[14,15]. However, enzyme digestion is time- and sample-consuming; therefore, these isomers tend to remain undetermined most of the time when they are studied using conventional MS methods.

Special MS methods for $N$-glycan structural determination have recently been reported to fill the gap of structural determination. For example, specific fragments produced from collision-induced dissociation (CID) are used as diagnostic ions to differentiate $N$-glycan isomers; typically, diagnostic ions are identified through comparison of existing MS/MS spectra of $N$-glycan isomer standards[16–21]. Reinhold et al. developed an MS method that uses multistage CID in an ion trap to differentiate high-mannose $N$-glycan isomers[11]. They identified the topologies of $N$-glycans based on ion compositions through the disassembly pathway of CID, and the linkage and branching were determined through spectral comparison with the spectra in an oligosaccharide database. Lin et al. used an ion cyclotron resonance mass spectrometer and electron excitation dissociation to identify the isomers of high-mannose $N$-glycans through machine learning from a set of tandem mass spectra of $N$-glycan standards[22]. Although these special MS methods can differentiate high-mannose $N$-glycan isomers, they require a database of mass spectra of glycan standards and oligosaccharides as a database and therefore, are limited to the $N$-glycan standards and oligosaccharides that are available.

For oligosaccharide structural determination, we recently developed a MS method, logically derived sequence (LODES) tandem mass spectrometry $(MS^n)$[23–27], that involves successive CID of intact oligosaccharide sodium (or lithium) adducts in an ion trap with the CID sequences derived from carbohydrate dissociation mechanisms[28–31]. The LODES/$MS^n$ does not require oligosaccharide standards and can be used to determine compositions, sequences, linkage positions, anomericity, and monosaccharide stereoisomers. In this study, we validated LODES/$MS^n$

by characterizing the structures of synthesized high-mannose and complex $N$-glycans and reported the step-by-step process of structural determination. Then, we applied LODES/$MS^n$ for the structural determination of $N$-glycans extracted from soybean, ovalbumin, and IgY.

## Results

**Dissociation mechanisms**. The sequences of CID used in LODES and the judgment propensities used in structural identification are derived from the dissociation mechanisms of oligosaccharide sodium adducts[28–31]. The mechanisms related to the structural determination of $N$-glycans used in this study are summarized as the following three propensities, where the notations of B, C, Y, Z, $^{0,2}A$, $^{0,3}A$, $^{0,4}A$, $^{0,2}X$, and $^{0,3}X$ are used according to the nomenclature of Domon and Costello[32].

*Propensity 1.* Dehydration mainly occurs on the sugar at the reducing end.

*Propensity 2.* Cross-ring dissociation mainly occurs on the sugar at the reducing end and follows retro-aldol reactions. The linkage of hexose at the reducing end is determined using the following six dissociation patterns derived from retro-aldol reactions.

(a) If the loss of neutrals m = 60, 90, and 120 from the cross-ring dissociation of $^{0,2}A$, $^{0,3}A$, and $^{0,4}A$, respectively, has the ratio of 5:3.5 ± 1:1 ± 0.5 (or 4:6 ± 1:1 ± 0.5), it represents a 1→6 linkage [or the 1→6 linkage generated by 1→3 glycosidic bond cleavage from the branch with (1→3, 1→6) linkages].

(b) If the loss of neutral m = 60 from the cross-ring dissociation of $^{0,2}A$ is much larger than the loss of neutrals m = 90 and 120, it represents a 1→4 linkage.

(c) If the loss of neutral m = 90 from the cross-ring dissociation of $^{0,3}X$ is much larger than the loss of neutrals m = 60 and 120, it represents a 1→3 linkage.

(d) If the loss of neutral m = 120 from the cross-ring dissociation of $^{0,2}X$ is much larger than the loss of neutrals m = 60 and 90, it represents a 1→2 linkage.

(e) If the loss of neutrals m = 60, 90, and 120 is much smaller than the cross-ring dissociation of $^{0,3}A$ and $^{0,3}X$, it represents a branch with 1→3 and 1→6 linkages.

(f) If the loss of neutrals m = 60, 90, and 120 is much smaller than the cross-ring dissociation of $^{0,2}X$ and $^{0,2}A$, it represents a branch with 1→2 and 1→4 linkages or with 1→2 and 1→6 linkages.

These dissociation patterns can be explained using retro-aldol reactions, as illustrated in the Supplementary Fig. S1. Our previous study[28] showed that the fragments produced from dehydration or crossed-ring dissociation at nonreducing end are typically less than 5% of the fragments produced from dehydration or crossed-ring dissociation at the reducing end, except few cases that they are up to 20%. Therefore, the aforementioned statements of "intensity of M fragment is much larger than that of N fragment" means "intensity of M fragment is at least five times larger than that of N fragment".

*Propensity 3.* Any glycosidic bond can be cleaved to produce B, C, Y, and Z ions. Cleavage of glycosidic bonds produces B, C, Y, Z ions are commonly observed in sodium adducts[33,34], although the intensities of B and Y ions are typically larger than that of C and Z ions.

## Validation of LODES/$MS^n$ using synthesized $N$-glycans

*$Man_4GlcNAc_2$ N-glycans.* Because similar CID sequences and judgment propensities for structural determination are used repeatedly for various $N$-glycans, we use $Man_4GlcNAc_2$ $N$-

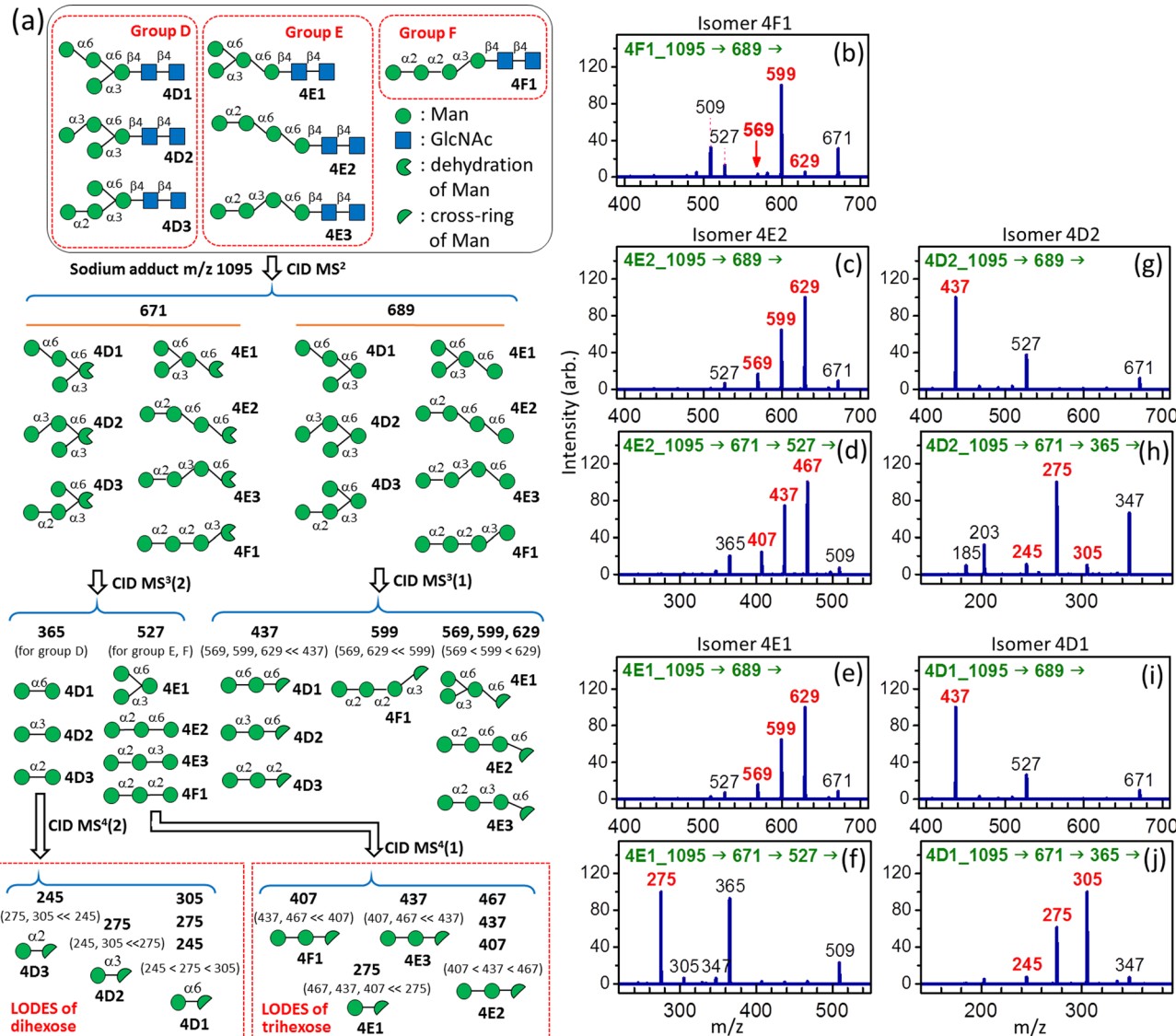

**Fig. 1 LODES and CID spectra of Man$_4$GlcNAc$_2$ N-glycans. a** LODES for Man$_4$GlcNAc$_2$ N-glycans; **b** CID spectrum of isomer **4F1**; **c, d** CID spectra of isomer **4E2**; **e, f** CID spectra of isomer **4E1**; **g, h** CID spectra of isomer **4D2**; **i, j** CID spectra of isomer **4D**1. The numbers (in green) in the horizontal line at the top of each CID spectrum represent the CID sequence. The number on the top of each peak in CID spectrum represents m/z value, which the structural decisive fragments are highlighted in red boldface. Details of the process of structural determination using these spectra are provided in the main text.

glycans, as they are the first example here, to demonstrate the detailed structural determination process. As illustrated in Fig. 1a, Man$_4$GlcNAc$_2$ N-glycans can have seven isomers, which are divided into three groups (groups D, E, and F). The first step in LODES/MS$^n$ is the generation of the C ion (tetra-mannose sodium adduct, m/z 689) produced from the cleavage of the Man-GlcNAc glycosidic bond, followed by the CID of this C ion [i.e., 1095→689→fragments; MS$^2$ and MS$^3$(1) in Fig. 1a]. Isomers can be classified into different groups from the mass spectrum of this CID sequence. N-glycans must belong to group F (i.e., 1→3 linkage at the reducing end of the C ion) if the intensity of fragment ion m/z 599 (loss of neutral m = 90 from the C ion m/z 689) is much higher than those of fragment ions m/z 629 (loss of neutral m = 60 from the C ion m/z 689) and m/z 569 (loss of neutral m = 120 from the C ion m/z 689), according to the dissociation mechanism in propensity 2(c). By contrast, N-glycans must belong to group E (i.e., 1→6 linkage at the reducing end of the C ion) if the relative intensities of fragment ions m/z 629, 599,

and 569 are approximately 5:3.5:1 [propensity 2(a)]. Finally, N-glycan must belong to group D (i.e., a branch with 1→3 and 1→6 linkages at the reducing end of the C ion) if the intensity of fragment ion m/z 437 (cross-ring dissociation of $^{0,3}$A and $^{0,3}$X of the C ion m/z 689) is much higher than those of fragment ions m/z 629, 599, and 569 [propensity 2(e)].

The second step is the differentiation of isomers in each group. The isomers in group E can be distinguished using the CID sequence: 1095→671→527→fragments [MS$^2$, MS$^3$(2), and MS$^4$(1) in Fig. 1a], where ion m/z 671 represents the B ion produced by breaking the Man-GlcNAc glycosidic bond and ion m/z 527 is the trimannose generated from the B ion. If the intensity of fragment ion m/z 437 (loss of neutral m = 90 from ion m/z 527) is much higher than those of fragment ions m/z 467 (loss of neutral m = 60 from ion m/z 527) and m/z 407 (loss of neutral m = 120 from ion m/z 527), then the N-glycan is **4E3** [propensity 2 (c)]. If the intensity ratio of fragment ions m/z 467, 437, and 407 is close to 5:3.5:1, then the N-glycan is **4E2** [propensity 2(a)]. If

the intensity of fragment ion $m/z$ 275 (cross-ring dissociation of $^{0,3}A$ and $^{0,3}X$ of ion $m/z$ 527) is much higher than those of fragment ions $m/z$ 467, 437, and 407, then the $N$-glycan is **4E1** [propensity 2(e)].

Differentiation of isomers in group D is conducted using the CID sequence: M→671→365→fragments [MS$^2$, MS$^3$(2), and MS$^4$(2) in Fig. 1a], where ion $m/z$ 365 is the mannose disaccharide generated from the B ion $m/z$ 671. If the intensity of fragment ion 275 (loss of neutral m = 90 from ion $m/z$ 365) is much higher than those of fragment ions $m/z$ 305 (loss of neutral m = 60 from ion $m/z$ 365) and $m/z$ 245 (loss of neutral m = 120 from ion $m/z$ 365), then the $N$-glycan is **4D2** [propensity 2(c)]. If the intensity ratio of ions $m/z$ 305, 275, and 245 is close to 5:3.5:1, then the $N$-glycan is **4D1** [propensity 2(a)]. If the intensity of ion $m/z$ 245 is much higher than those of ions $m/z$ 305 and 275, then the $N$-glycan is **4D3** [propensity 2 (d)].

To validate LODES/MS$^n$, it was applied to the structural determination of all possible Man$_4$GlcNAc$_2$ glycan isomers from chemical synthesis; the validation is shown in Fig. 1b–j (five isomers) and Supplementary Fig. S2 (two isomers). A step-by-step procedure using the aforementioned guideline for the structural determination of isomers in Fig. 1b–j is provided as follows.

**Isomer 4F1**. The intensity of ion $m/z$ 599 is much higher than the intensities of ions $m/z$ 629, 569, and 437 in Fig. 1b, indicating that the isomer belongs to group F [propensity 2(c)]. Because only one isomer is present in this group, it must be 4F1.

**Isomer 4E2**. The intensities of ions $m/z$ 629, 599, and 569 in Fig. 1c are much higher than the intensity of ion $m/z$ 437, and they are close to the ratio of 5:3.5:1, indicating that the isomer belongs to group E [propensity 2(a)]. The intensities of ions $m/z$ 407,437, and 467 in Fig. 1d are much higher than the intensity of ion $m/z$ 275, and they are close to the ratio of 5:3.5:1, indicating that the isomer is 4E2 [propensity 2(a)].

**Isomer 4E1**. The intensities of ions $m/z$ 629, 599, and 569 in Fig. 1e are much higher than the intensity of ion $m/z$ 437, and they are close to the ratio of 5:3.5:1, indicating that the isomer belongs to group E [propensity 2(a)]. The intensity of ion $m/z$ 275 is much higher than the intensities of ions $m/z$ 407, 437, and 467 in Fig. 1f, indicating that the isomer is 4E1 [propensity 2(e)].

**Isomer 4D2**. The intensity of ion $m/z$ 437 is much higher than the intensities of ions $m/z$ 629, 599, and 569 in Fig. 1g, indicating that the isomer belongs to group D [propensity 2(e)]. The intensity of ion $m/z$ 275 is much larger than those of ions $m/z$ 245 and 305 in Fig. 1h, indicating that the isomer is 4D2 [propensity 2 (c)].

**Isomer 4D1**. The intensity of ion $m/z$ 437 is much higher than the intensities of ions $m/z$ 629, 599, and 569 in Fig. 1i, indicating that the isomer belongs to group D [propensity 2(e)]. The intensity ratio of ions $m/z$ 305, 275, and 245 is close to 5:3.5:1 in Fig. 1j, indicating that the isomer is 4D1 [propensity 2(a)].

*Asymmetrical biantennary complex* N-*glycans*. In asymmetrical biantennary complex $N$-glycans, one arm (Manα1-6 arm or Manα1-3 arm) is longer than the other. The structural decisive fragment is the ion that consists of all the monosaccharides of the longer arm. The generation of this fragment requires the successive cleavage of at least two glycosidic bonds: the glycosidic bond of Man-GlcNAc (generation of C ion) and the glycosidic bond of α-Man-(1→3)-Man or α-Man-(1→6)-Man, which is connected to the shorter arm (generation of Y ion). The cleavage of these two bonds can occur in any sequence; therefore, the structural decisive fragment can be generated through various CID sequences. Here, isomers **3-G1** and **6-G1** are used as examples to demonstrate the structural determination of

asymmetrical biantennary complex $N$-glycans [Fig. 2a]. Isomers **3-G1** and **6-G1** can be distinguished using the CID spectrum of the structural decisive fragment ion $m/z$ 730 [MS$^4$(1) in Fig. 2a]. If the intensity ratio of fragment ions from the loss of neutrals m = 60, 90, and 120 is close to 4:6:1 in the CID spectrum of ion $m/z$ 730, then the isomer is **6-G1** [propensity 2(a)]. By contrast, if the intensity of the fragment ion from the loss of neutral m = 90 is much higher than those of the ions from the loss of neutrals m = 60 and 120 in the CID spectrum of ion $m/z$ 730, then the isomer is **3-G1** [propensity 2(c)]. In the validation of LODES/MS$^n$ for the structural determination of the **6-G1** and **3-G1** isomers [Fig. 2b–d], the precursor is a double-charged ion with two Na$^+$ ions. Two CID sequences are used to generate the structural decisive fragment ion for the structural determination of the **6-G1** isomer, as shown in the CID sequences in Fig. 2b, c, and the same structure is identified based on these sequences. This demonstrates that LODES/MS$^n$ can derive from multiple CID sequences, providing a crosscheck for structural identification.

The analogous CID sequences (generating the structural decisive ion) and the same judgment propensities can be applied to any asymmetrical biantennary complex $N$-glycan.

*Triantennary complex* N-*glycans*. LODES for the structural determination of triantennary complex $N$-glycans required the C ion with a branch (i.e., two antennaries) produced through breakage of the glycosidic bond of α-Man-(1→6)-Man in the Manα1-6 arm or of α-Man-(1→3)-Man in the Manα1-3 arm. This C ion is a structural decisive fragment that can be used for all types of triantennary $N$-glycans. For the $N$-glycans containing sialic acid, structural determination is conducted in two steps. The first step involves the elimination of sialic acid, followed by the determination of the $N$-glycan structure without sialic acid. The second step is the determination of the glycosidic bond of sialic acid.

Analogous to the LODES of biantennary $N$-glycans, there are multiple CID sequences for the structural determination of triantennary $N$-glycans, and these sequences can be used for crosschecking the results. LODES for the triantennary $N$-glycans is shown in Fig. 3a. In step 1, ions $m/z$ = 609 or 771 produced in MS$^4$(1) are the structural decisive fragments. Observation of ion $m/z$ 771 indicates that the isomers belong to group D or E; otherwise, the isomers belong to group F. Further differentiation of isomers in each group is conducted using subsequent CID of ion $m/z$ 771 or 609 [Fig. 3a]. In step 2, the glycosidic bond of sialic acid is determined. Because sialic acid is easily eliminated during the CID process, a special CID sequence is used. First, the fragment resulting from $CO_2$ elimination (from the carboxyl group of sialic acid) of the precursor ion is selected, followed by the CID to generate sialylgalactose (in which $CO_2$ is eliminated and is denoted as sialylgalactose without $CO_2$). Then, for structural determination, the CID spectrum of sialylgalactose without $CO_2$ generated from $N$-glycans is compared with the CID spectra of sialylgalactose without $CO_2$ in a database (Fig. 3b, c). Detailed measurements of the CID spectra of the sialylgalactose without $CO_2$ in the database are presented in the section of Supplementary Note 1 and Supplementary Fig. S3.

A step-by-step description of the procedures for the structural determination of three isomers using the aforementioned guidelines is provided as follows.

**Isomer α-2-3 E1**. The ion $m/z$ 771 in Fig. 3d indicates that the isomer belongs to group D or E. The intensity of ion $m/z$ 448 is much higher than the intensities of ions $m/z$ 478 and 508 in the CID spectrum of ion $m/z$ 568 (Fig. 3e), indicating that the isomer belongs to group E [propensity 2(d)]. The intensity ratio of ions $m/z$ 346 and 316 is close to 5:3.5 in Fig. 3f, indicating that the isomer

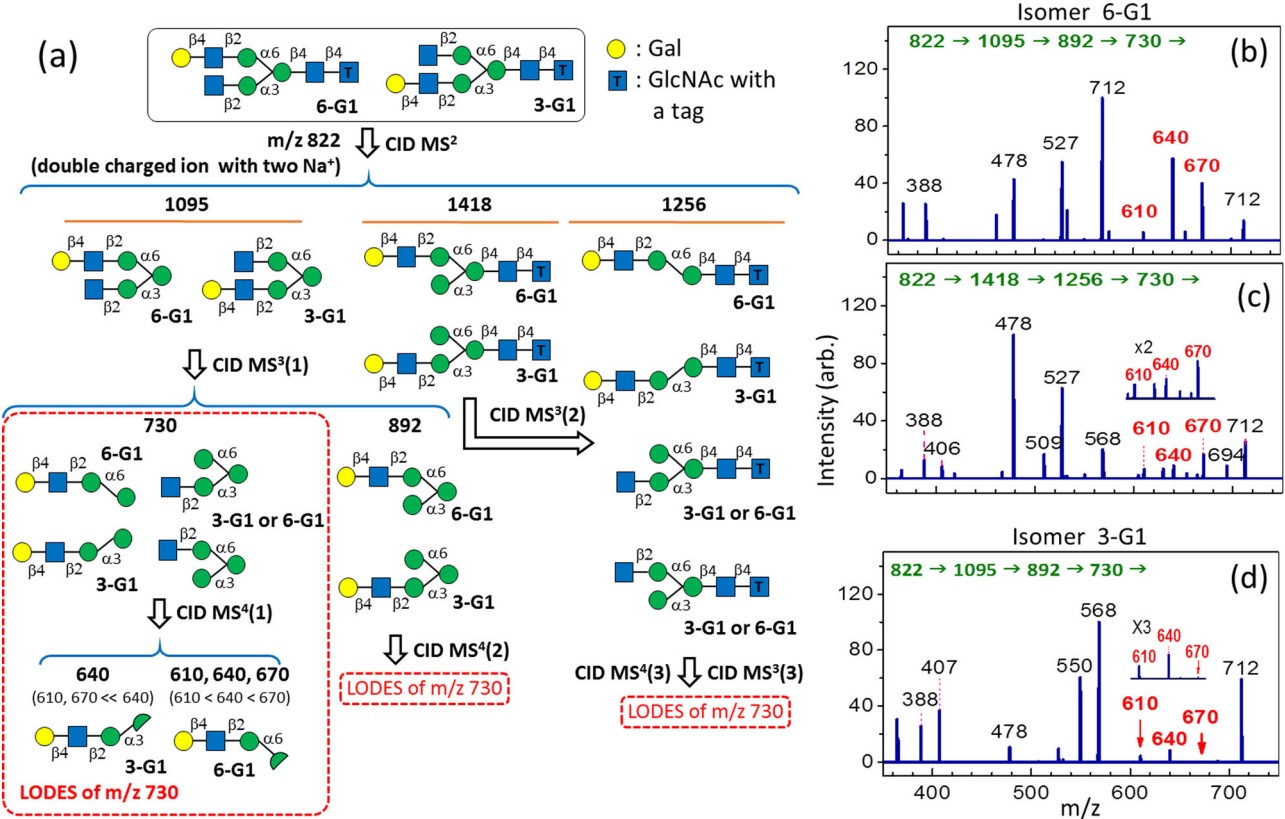

**Fig. 2 LODES and CID spectra of asymmetrical biantennary *N*-glycans. a** LODES for the complex *N*-glycans **3-G1** and **6-G1** with the tag of 2-aminobenzamide acid (2-AB) at the reducing end. **b**, **c** CID spectra of isomer **6-G1** through two CID sequences. Ion *m/z* 730 in (**b**) is produced by the α-1→3 glycosidic bond cleavage from the branch with (α-1→3, α-1→6) linkages, and the intensity ratio of ions *m/z* 670, 640, and 610 [loss of neutrals (m = 60, 90, and 120) from ion *m/z* 730, respectively] is close to 4:6:1, indicating that the isomer is **6-G1** [propensity 2(a)]. Ion *m/z* 730 in (**c**) is produced by the cleavage of β-Man-(1→4)-GlcNAc glycosidic bond, and the intensity ratio of ions *m/z* 670, 640, and 610 is close to 5:3.5:1, indicating that the isomer is **6-G1** [propensity 2(a)]. **d** CID spectrum of isomer **3-G1**. The intensity of ion *m/z* 640 is much higher than the intensities of ions *m/z* 670 and 610, indicating the isomer is **3-G1** [propensity 2(c)]. The numbers (in green) in the horizontal line at the top of each CID spectrum represent the CID sequence. The number on the top of each peak in CID spectrum represents *m/z* value, which the structural decisive fragments are highlighted in red boldface.

is **E1** [propensity 2(a)]. Notably, ion *m/z* 406 in Fig. 3f includes disaccharides GlcNAc-(1→6)-Man (or GlcNAc-(1→4)-Man) and GlcNAc-(1→2)-Man, as indicated by MS⁶(3) in Fig. 3a. Ion *m/z* 286 produced from cross-ring dissociation of GlcNAc-(1→2)-Man disturbs the intensity ratio of ions *m/z* 346, 316, and 286 produced from the cross-ring dissociation of GlcNAc-(1→6)-Man [or GlcNAc-(1→4)-Man]. Thus, the intensity of ion *m/z* 286 in Fig. 3f is not included in the comparison of the intensities of ions *m/z* 346 and 316. Finally, a comparison of Fig. 3g with Fig. 3b, c indicates that the glycosidic bond of sialic acid is α-2→3.

**Isomer α-2-3 F2**. The intensity of ion *m/z* 609 is much higher than that of ion *m/z* 771 in Fig. 3h, which indicates that the isomer belongs to group F, according to MS⁴(1) in Fig. 3a. The intensity of ion *m/z* 346 is much larger than that of ion *m/z* 316 in Fig. 3i, indicating the isomer is **F2** [propensity 2(b)]. Analogous to the previous example, ion *m/z* 406 in Fig. 3i includes disaccharides GlcNAc-(1→6)-Man and GlcNAc-(1→2)-Man; thus, the intensity of ion *m/z* 286 in Fig. 3i is not included in the comparison of the intensities of ions *m/z* 346 and 316. A comparison of Fig. 3j with Fig. 3b, c indicates that the glycosidic bond of sialic acid is α-2→3.

**Isomer α-2-6 D1**. The ion *m/z* 771 observed in Fig. 3k indicates the isomer belongs to group D or E. The intensity ratio of ions *m/z* 508, 478, and 448 is near 5:3.5:1 in Fig. 3l, indicating the isomer is **D1** [propensity 2(a)]. A comparison of Fig. 3m with Fig. 3b, c indicates that the glycosidic bond of sialic acid is α-2→6.

*Tetra-antenny complex* N*-glycans*. LODES for the structural determination of tetra-antennary *N*-glycans is very similar to that of triantennary *N*-glycans. Two tetra-antennary *N*-glycan isomers, α-2-3 **E2** and α-2-6 **D2**, were studied in this work. LODES for the tetra-antennary *N*-glycans is shown in Fig. 4a, and the corresponding CID spectra used to validate this method are displayed in Fig. 4b–g.

**N-glycans extracted from biological samples**. Soybean proteins, ovalbumin, and antibody IgY from hen eggs are natural products frequently used to extract *N*-glycans for further applications[35–38]. However, most *N*-glycan isomers extracted from these materials remain unidentified. One major *N*-glycan in soybean protein is Hex₈GlcNAc₂ *N*-glycan. We found only two peaks in the chromatogram [Fig. 5a] after separating by porous graphitic carbon (PGC) column, the last column in multidimensional HPLC separation (see details in Experimental Method). PGC column is known for separating anomeric isomers, so the two peaks in Fig. 5a might be the alpha and beta anomers of the same Hex₈GlcNAc₂ *N*-glycan isomer. To clarify this, the eluents separated from the PGC column were collected in fraction every 30 s so that compound corresponding to each peak in Fig. 5a was collected into different test tubes. The same collection was repeated for ten times which the eluents of the same retention time were combined in the same test tube. Then, these fractionated

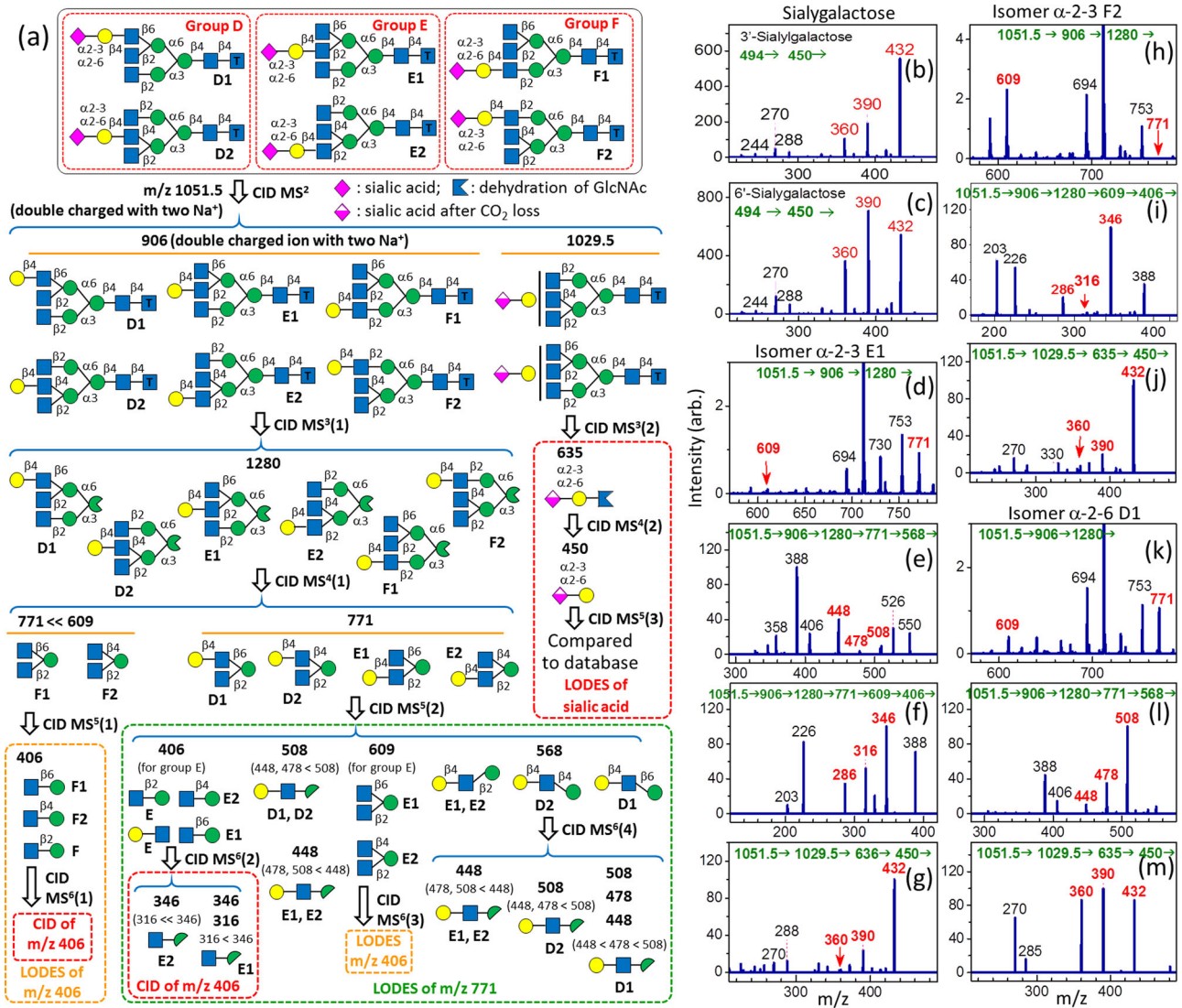

**Fig. 3 LODES and CID spectra of triantennary *N*-glycans. a** LODES for triantennary *N*-glycans with the tag $(CH_2)_5NH_2$ at reducing end; **b, c** CID spectrum of 3′-sialylgalactose and 6′-sialylgalactose. **d–g** CID spectra of isomer **α-2-3 E1**; **h–j** CID spectra of isomer **α-2-3 F1**; **k–m** CID spectra of isomer **α-2-6 D1**. The numbers (in green) in the horizontal line at the top of each CID spectrum represent the CID sequence. The number on the top of each peak in CID spectrum represents *m/z* value, which the structural decisive fragments are highlighted in red boldface.

eluents were stored at room temperature for more than 6 h before they were concentrated and reinjected into the same PGC column (with the same separation gradient) individually. If the two peaks in Fig. 5a belong to one isomer and they are only different by the anomericity at the reducing end, the reinjection of the fractionated eluents into the same PGC column would show two peaks in chromatogram, and the relative intensities and the retention times of these two peaks must be the same as that in Fig. 5a. This is because the anomers undergo mutarotation, i.e., change from α (or β) configuration to β (or α) configuration, in solution and this mutarotation typically takes only about 30 min–2 h at room temperature. Although each fraction contains only one anomer when it was collected right after the PGC separation, the anomer undergoes mutarotation during the 6-h storage time. As a result, there are two anomers in each collected fraction and these two anomers reach equilibrium. The PGC chromatograms of these fractionated eluents, as illustrated in Fig. 5b, indeed shows two peaks at the same retention time with the same relative intensities as that in Fig. 5a. Tube 10 (fraction collected at 21.5–22 min) and tube 14 (fraction collected at 23.5–24 min) have the largest

intensity among all the collected fractions, as they correspond to the fractions collected at the retention times of two peaks in the first PGC chromatogram [Fig. 5a], respectively. Consequently, we can conclude that the two peaks in the first PGC chromatogram [Fig. 5a] belong to one isomer and they are only different by the anomericity at the reducing end. The structure was identified as $8A_{1,2,3}B_{1,3}$. Details of the CID spectra for the structural determination are presented in the Supplementary Fig. S4.

To demonstrate that LODES/MS[n] work for the reduced *N*-glycans, the *N*-glycans extracted from hen egg ovalbumin were reduced before structural analysis. Two peaks, of $Man_4GlcNAc_2$ *N*-glycans, representing two isomers, were found [Fig. 5b]. They were identified as $4A_1$ and $4A_2$ [i.e., isomers **4D1** and **4D2** in Fig. 1a]. Details of the CID spectra for the structural determination are presented in the Supplementary Fig. S5.

Typically, *N*-glycans are generated from the sequential removal of Glc residues from $Glc_3Man_9GlcNAc_2$ by α-glucosidases I and II, yielding $Man_9GlcNAc_2$; this is followed by the removal of Man by mannosidase. However, the removal of Glc may be incomplete, resulting in glucosylated high-mannose *N*-glycans.

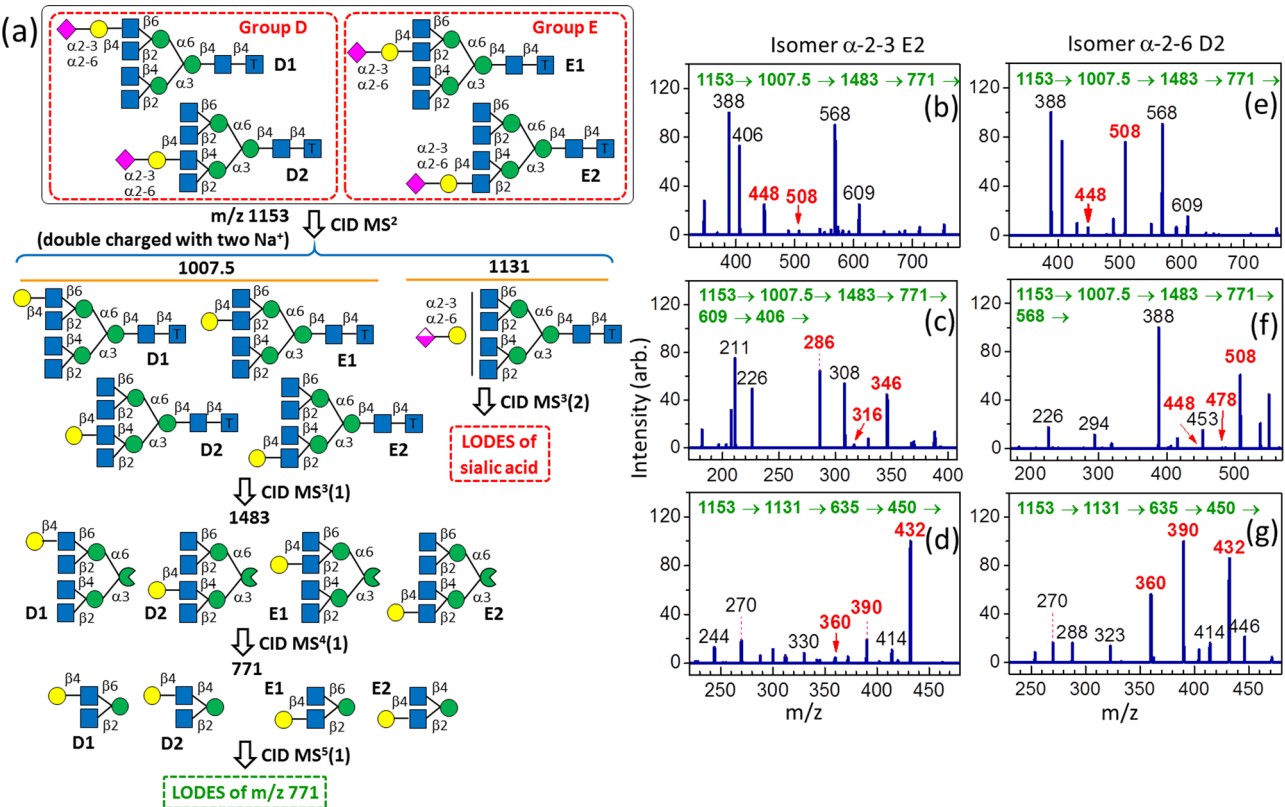

**Fig. 4 LODES and CID spectra of tetra-antennary *N*-glycans. a** LODES for tetra-antennary *N*-glycans with the tag of $(CH_2)_5NH_2$ at the reducing end. The LODES of *m/z* 771 (denoted by the dashed green line) and the LODES of sialic acid (denoted by the dashed red line) in (**a**) are analogous to that in Fig. 3a. **b–d** CID spectra of isomer **α-2-3 E2**. Intensity (ion *m/z* 448) > intensity (ion *m/z* 508) in (**b**) indicates that the isomer belongs to group E [propensity 2 (f)]. Intensity (ion *m/z* 346) > intensity (ion *m/z* 316) in (**c**) indicates that the isomer belongs to group E2 [propensity 2(b)]. The ion *m/z* 406 in (**c**) includes disaccharides GlcNAc-(1→4)-Man and GlcNAc-(1→2)-Man; thus, the intensity of ion *m/z* 286 in (**c**) is not included in the comparison of the intensities of ions *m/z* 346 and 316. **e–g** CID spectra of isomer **α-2-6 D2**. Intensity (ion *m/z* 448) < intensity (ion *m/z* 508) in (**e**) indicates that the isomer belongs to group D [propensity 2(f)]. Intensity (ion *m/z* 508) > intensity (ion *m/z* 478) and intensity (*m/z* = 448) in (**f**) indicates that the isomer is D2 (propensity 2(b)). A comparison of (**d**) and (**g**) with Fig. 3b, c suggests the glycosidic bond of sialic acid is α-2→3 and α-2→6, respectively. The numbers (in green) in the horizontal line at the top of each CID spectrum represent the CID sequence. The number on the top of each peak in CID spectrum represents *m/z* value, which the structural decisive fragments are highlighted in red boldface.

Glucosylated high-mannose *N*-glycans have been found from hen egg yolk IgY[5,39]. We found only one major isomer of $Hex_{10}GlcNAc_2$ [Fig. 6b]. The structure was identified as isomer $10A_{1,2,3}B_{1,2,3}C_3$ [i.e., isomer **10D** in Fig. 6a] using LODES/MS^n and the mass spectra presented in Fig. 6a, c, d. Complete LODES of $Hex_{10}GlcNAc_2$ and detailed identification that these two peaks belong to one isomer are illustrated in Supplementary Figs. S6 and S7, respectively.

## Discussion

We demonstrated that the structures of *N*-glycan isomers can be determined using only 1–4 LODES selected CID spectra. Typically, less than 100 pmol is required for all necessary CID spectrum measurements. Notably, LODES/MS^n can be applied to intact *N*-glycans. This method does not require permethylation, reduction, or labeling at the reducing end of *N*-glycans, although it can be used for *N*-glycans reduced or labeled at the reducing end. Using intact *N*-glycans eliminates the effort of reduction and labeling of *N*-glycans and greatly reduces the loss of sample during sample pretreatment. For LODES/MS^n, the mass spectra of small oligosaccharides or *N*-glycan standards as the database are not required; therefore, this method is not limited to available oligosaccharides and *N*-glycan standards. LODES/MS^n can be applied to high-mannose, hybrid, and complex *N*-glycans, including *N*-glycans produced from typical biosynthetic pathways

and *N*-glycans degraded from larger *N*-glycans by any enzyme or acid hydrolysis. The same dissociation mechanisms (propensities 1–3) can be used to derive LODES for these *N*-glycans.

LODES/MS^n is also an efficient method for developing a *N*-glycan MS library. The conventional approach for the development of an MS library for *N*-glycan structural identification requires the CID spectra of MS^n from *N*-glycan standards. Moreover, identification of the fragments that have the greatest difference between isomers in the CID spectra in MS^n is crucial for differentiating isomers. LODES provides a guideline for the selection of these "diagnostic fragments" to differentiate isomers, which greatly reduces the effort required for the development of an MS library. The entire LODES can be coded in the program for automatic *N*-glycan structural determination, providing considerable advantages for carbohydrate structural determination.

## Methods

### Materials

*Sources of materials.* High-mannose Man₄ *N*-glycans were purchased from Omicron Biochemicals, Inc. (South Bend, IN, USA), and biantennary *N*-glycans (3-G1 and 6-G1) were purchased from TCI (Tokyo Chemical Industry Co., Ltd. Tokyo, Japan). Tri and tetra-antennary complex *N*-glycans were synthesized in our laboratory. Details of the synthesis are provided in a previous study[40]. The NMR spectra and assignment of the *N*-glycans synthesized in our laboratory are illustrated in Supplementary Fig. S8–S17. Hen eggs purchased from a local market were used to extract IgY, soybean proteins were purchased from the Hut.com Ltd.

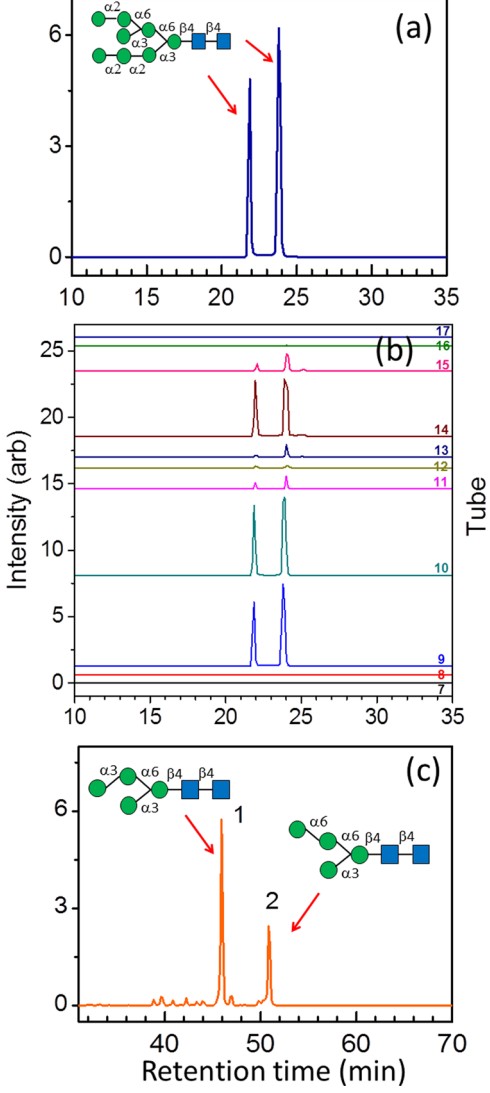

**Fig. 5 Chromatograms and structural assignments of Hex$_8$GlcNAc$_2$ N-glycan from soybean and Man$_4$GlcNAc$_2$ N-glycans from hen egg ovalbumin. a** Chromatograms (total intensity of fragments from ion m/z 1743) of Hex$_8$GlcNAc$_2$ N-glycan extracted from soybean proteins. This chromatogram shows the last separation in two-dimensional high-performance liquid chromatography (HPLC). **b** Chromatograms of the eluents collected from the chromatogram in (**a**). The eluents corresponding to the chromatogram in (**a**) were collected every 30 s. After repeating the collection of ten times, the eluents were stored in collecting tubes at room temperature for 6 h. Then the eluents were concentrated and reinjected into the same HPLC separately. The same retention time and relative intensities of the two peaks in chromatogram (**b**) are the same as that in chromatogram (**a**), representing they represent the α and β anomeric configurations of the sugar at the reducing end of the same isomer. **c** Chromatogram (total intensity of fragments from ion m/z 1097) of Man$_4$GlcNAc$_2$ N-glycans extracted from hen egg ovalbumin. This chromatogram shows the last separation in four-dimensional HPLC. The reduction of Man$_4$-N-glycans was performed to demonstrate that LODES/MS$^n$ can be applied to N-glycans after reduction. Two peaks in the chromatogram represent two isomers. The details of HPLC conditions are described in Methods and the mass spectra used in LODES/MS$^n$ for structural determination are presented in Supplementary Information Figs. S4 and S5.

(Manchester, UK), and hen egg ovalbumin was purchased from Grade V, Merck, Darmstadt, Germany.

*Extraction of IgY from hen egg yolk.* Hen yolk IgY purification kits (Gallus Immunotech Inc., Nanterre, France) were used in this study. The yolk was first separated from the egg white using an egg separator and was then rinsed with distilled water. The content of the yolk was separated from the yolk membrane by using a pipette. Cold reagent A (4 °C) from the purification kit (at a volume five times larger than that of the yolk) was slowly added to the yolk. The mixture was stirred slowly for ~15 min and then kept at 4 °C for 6 h. The solution was then centrifuged at 4000×g for 20 min at 4 °C, and the supernatant was collected. The supernatant was mixed with an equal volume of cold reagent B (4 °C) from the purification kit, stirred gently for 2 min at 4 °C, and then kept at 4 °C for 6 h. The solution was centrifugated at 4000×g for 20 min at 4 °C, and IgY was collected from the precipitate.

*N-glycans released from IgY and soybean proteins.* N-glycans were released from IgY and soybean proteins through an ammonia-catalyzed reaction described in a previous study[41]. In brief, IgY (or soybean protein) was dissolved in a 25% ammonia aqueous solution for a 16-h reaction at 60 °C. After the reaction, the ammonia in the solution was removed using a rotary evaporator, and proteins were removed through ethanol precipitation. The released N-glycans were purified using a C18 cartridge (Sep-Pak C18, Waters, Milford, MA, USA) for the removal of residual proteins, followed by the removal of potential contaminants and salt using size exclusion chromatography (TOYOPEARL HW-40F, Tosoh Bioscience GmbH, Griesheim, Germany).

Two-dimensional high-performance liquid chromatography (HPLC) separation was used to separate N-glycans. The first dimension was HPLC separation using a TSKgel amide-80 column (150 mm × 2.0 mm, particle size of 5 μm, Tosoh Bioscience GmbH, Griesheim, Germany). The fractions collected from the first HPLC were input into the second HPLC separation using a Hypercarb column (2.1 mm × 100 mm, particle size of 3 μm, Thermo Fisher Scientific, Waltham, MA, USA). The mobile phases used in HPLC were deionized water (solution A) and HPLC-grade acetonitrile (solution B). The HPLC conditions for the TSKgel Amide-80 column were as follows: The flow rate was 0.2 mL/min, the gradient was changed linearly from A = 35% and B = 65% at t = 0 to A = 45% and B = 55% at t = 50 min. The HPLC conditions for the Hypercarb column were as follows: The flow rate was 0.2 mL/min; the gradient was changed linearly from A = 92%, B = 8% at t = 0 to A = 82%, B = 18% at t = 30 min. A CM 5000 series HPLC (Chromaster, Hitachi, Chiyoda-ku, Tokyo, Japan) and fraction collector (FC204, Gilson, Middleton, WI, USA) were used for separation and fraction collection.

*N-glycans from hen egg ovalbumin.* N-linked glycans were released from hen egg ovalbumin (Grade V, Merck, Darmstadt, Germany) using PNGase F (New England Biolabs, Ipswich, MA, USA) in a solution consisting of 50 mM sodium phosphate (pH 7.5) with 24-h incubation at 37 °C. The released glycans were purified through ethanol precipitation to remove proteins, followed by solid-phase extraction using a C18 cartridge to further remove residual proteins; a NPGC cartridge (Extract-Clean SPE Carbo, Grace, Columbia, MD, USA) was used to remove salt. These N-glycans were then reduced in a 50 mM KOH aqueous solution consisting of 1 M NaBH$_4$ for 3 h at 50 °C; subsequently, glacial acetic acid was added to quench reduction and neutralized the solution to pH 7.0–7.5. The reduced glycans were then desalted and extracted using the NPGC cartridge. Man$_4$ N-glycan was separated from other N-glycans through four-dimensional HPLC separation. The TSKgel SuperOligoPW column (particle size of 3 μm, 6.0 mm × 150 mm, Tosoh, Griesheim, Germany) was used in the first dimension HPLC, followed by a GlycanPac AXH-1 column (particle size of 3 μm, 4.6 mm × 150 mm, Thermo Fisher Scientific) was used in the second dimension, an Accucore 150 Amide column (particle size of 2.6 μm, 2.1 mm × 150 mm, Thermo Fisher Scientific) in the third dimension, and an Hypercarb column (particle size of 3 μm, 2.1 mm × 150 mm, Thermo Fisher Scientific) in the fourth dimension. The HPLC system and fraction collector used in soybean proteins were used here for separation and fraction collection, respectively. Deionized water (solution A) and HPLC-grade acetonitrile (solution B) were used in HPLC as the mobile phases. The gradients of the mobile phase were as follows: for the TSKgel SuperOligoPW column, the flow rate was 0.14 mL/min, using isocratic gradient of A = 100% at t = 0 min to t = 90 min, and performed in ambient temperature; for the GlycanPac AXH-1 column, the experiment was performed at 50 °C, the flow rate was 0.4 mL/min, the gradient was changed linearly from A = 24% and B = 76% at t = 0 to A = 25% and B = 75% at t = 9 min; subsequently, the flow rate was changed to 0.2 mL/min and the gradient was changed linearly from A = 25% and B = 75% at t = 9 min to A = 32% and B = 68% at t = 72 min. For the Accucore 150 Amide column, the experiment was performed at 50 °C, the flow rate was 0.15 mL/min and the gradient was changed linearly from A = 25% and B = 75% at t = 0 min to A = 50% and B = 50% at t = 75 min. For the Hypercarb column, the experiment was performed at 50 °C, the flow rate was 0.15 mL/min and the gradient was changed linearly from A = 100% and B = 0% at t = 0 min to A = 75% and B = 25% at t = 75 min.

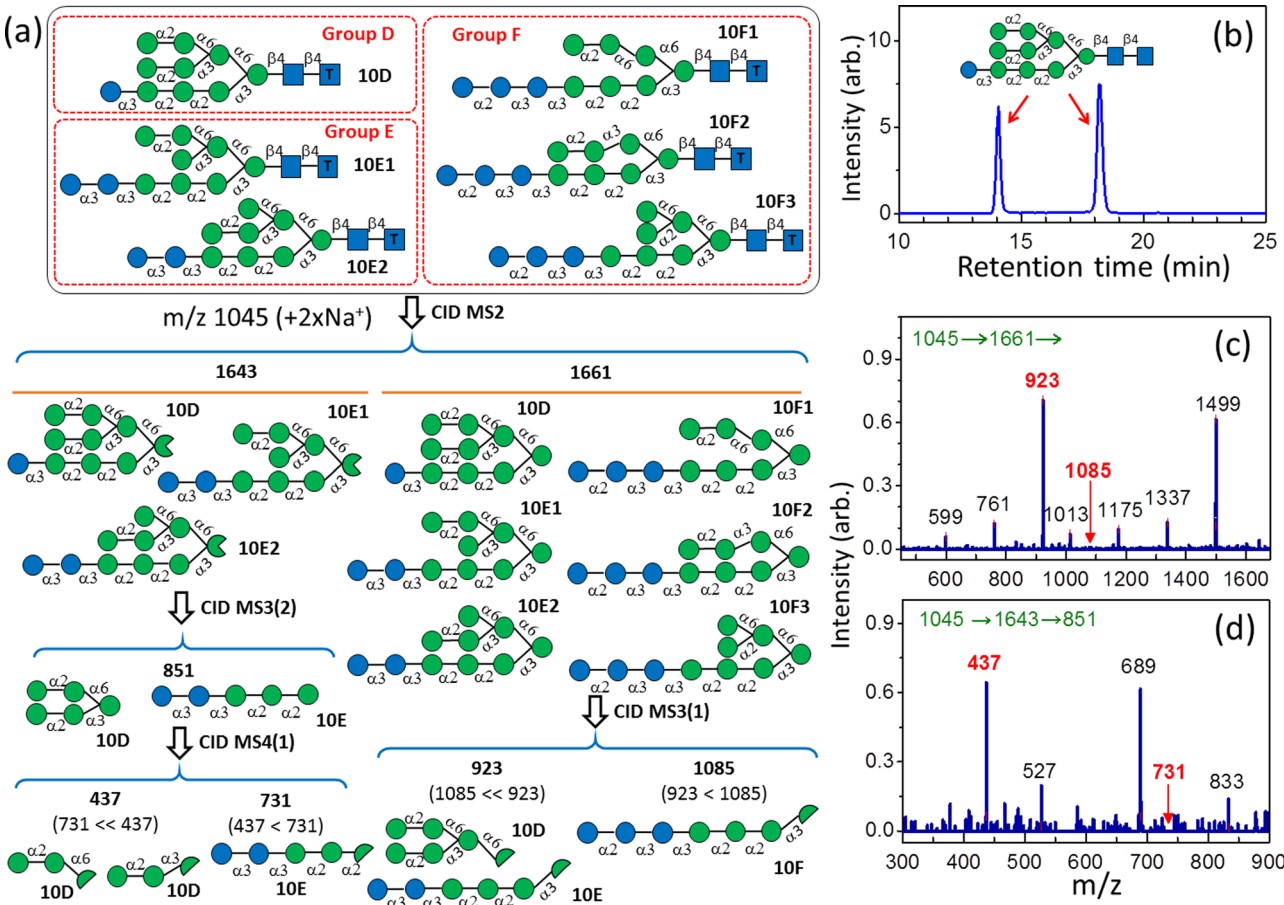

**Fig. 6 LODES, chromatograms, and CID spectra of Hex$_{10}$GlcNAc$_2$ N-glycans from IgY. a** LODES for the structural determination of Hex$_{10}$GlcNAc$_2$ N-glycans. Only the LODES used for identifying isomer **10D** is illustrated. The complete LODES of Hex$_{10}$GlcNAc$_2$ N-glycans is illustrated in the Supplementary Fig. S6. **b** Chromatogram (selective ion monitoring of ion m/z 1045) of Hex$_{10}$GlcNAc$_2$ N-glycan extracted from IgY. The chromatogram denotes the last separation in two-dimensional HPLC. Two peaks in the chromatogram represent the α and β anomeric configurations of the sugar at the reducing end of the same isomer. **c, d** CID mass spectra used to identify the structure. The intensity of ion m/z 923 is much higher than that of ion m/z 1085 in (**b**), indicating the Hex$_{10}$GlcNAc$_2$ belongs to groups D or E, according to MS$^3$(1) in (**a**). The intensity of ion m/z 437 is much higher than that of ion m/z 731 in (**c**), indicating Hex$_{10}$GlcNAc$_2$ is isomer **10D** according to MS$^4$(1) in (**a**). The numbers (in green) in the horizontal line at the top of each CID spectrum represent the CID sequence. The number on the top of each peak in CID spectrum represents m/z value, which the structural decisive fragments are highlighted in red boldface.

**Mass spectrometry**. LODES uses low energy CID (He as collision gas) and multistage tandem MS for structural identification. Any mass spectrometers with these capabilities in principle can be used. We used LTQ XL linear ion trap mass spectrometer (Thermo Fisher Scientific Inc., Waltham, MA, USA) in this study simply because this instrument is readily available in our laboratory. We found that CID spectra (i.e., relative intensities of fragments) do not change significantly in the normalized collision energy ranged from 25 to 100% in the instrument we used. For most of the CID spectra shown in this study, the normalized collision energies of 30–40% were used.

*Nanoelectrospray-mass spectrometry.* Samples of synthesized N-glycans were prepared in a 50:50 (vol/vol) water/methanol mixture at a concentration of $5 \times 10^{-5}$ M with NaCl ($5 \times 10^{-5}$ M). A nanoelectrospray ionization instrument coupled to a linear ion trap mass spectrometer (LTQ XL, Thermo Fisher Scientific) with a Nanospray Flex housing (Thermo Fisher Scientific) was used for MS. Specifically, 2 µL (the minimum volume for the emitter of ESI) of each sample was loaded into a borosilicate glass ESI emitter, which was produced in our laboratory using a P-97 (Sutter Instruments, Novato, CA, USA) Flaming/Brown micropipette puller. The emission was monitored using a DinoLite Premier digital microscope (model AM4113TL, AnMo Electronics, Taipei, Taiwan). The volume loaded can last for more than 2 h. The measurement of each spectrum took 1–20 min, depending on the signal-to-noise ratio. The ESI source voltage was 1.5 kV. In the mass spectrometer, the capillary voltage was 130 V, heated capillary temperature was 120 °C, and tube-lens voltage was 230 V. Helium gas was used as a buffer gas for the ion trap as well as a collision gas in CID. The pressure of He gas at the output of regulator connected to gas cylinder was set at the specification (40 psi). The pressure measured by the ion gauge in the vacuum chamber of mass spectrometer

was $0.9 \times 10^{-5}$ Torr. The MS$^n$ experiments were performed at an activation Q value of 0.25, an activation time of 30 ms, normalized collision energy 30–40%. The number of ions was regulated by injection time (10–20 ms) or automatic gain control ($1 \times 10^5$ for full scan, and $1 \times 10^4$ for MS$^n$). The precursor ion isolation width was set to 1 u.

*HPLC-electrospray-mass spectrometry.* The chromatograms of N-glycans extracted from biological samples [Figs. 5 and 6b] were measured by using a heated electrospray ionization (HESI-II) probe with an Ion Max housing and a linear ion trap mass spectrometer (LTQ XL, Thermo Fisher Scientific, Waltham, MA USA) coupled with an HPLC system (Dionex Ultimate 3000, Thermo Fisher Scientific). The entire HPLC and mass spectrometer system is controlled by using Dinoex Chromatography MS Link 2.14, Chromeleon Version 6.80 SR13, LTQ Tune Plus Version 2.7.0.1103 SP1, and Thermo Xcalibur 2.2 SP1.48 software from Thermo Fisher Scientific. The settings of mass spectrometer are the same as that in nanoelectrospray-MS, except the ion spray voltage was 4.00 kV, the transfer capillary temperature was 280 °C, the capillary voltage was 80 V, and the tube lens voltage was 150 V.

For the chromatogram of Hex$_8$GlcNAc$_2$ N-glycan extracted from soybean [Fig. 5a], the HPLC conditions are follows. Hypercarb column (3 µm, 2.1 × 100 mm Thermo Fisher Scientific) was used at room temperature, mobile phase gradient was changed linearly from water (100%) at $t = 0$ to water (82%) and ACN (18%) at $t = 30$ min, and the flow rate was 0.2 mL/min and NaCl ($1 \times 10^{-4}$ M in 80% MeOH aqueous solution) was added by post-column infusion.

For the chromatogram of Man$_4$GlcNAc$_2$ N-glycans extracted from hen egg ovalbumin [Fig. 5b], the HPLC conditions are follows. Hypercarb column (3 µm, 2.1 × 150 mm, Thermo Scientific) was used, temperature was 50 °C, mobile phase

gradient was changed linearly from water (100%) at $t = 0$ to water (75%) and ACN (25%) at $t = 75$ min, and the flow rate was 0.15 mL/min. The output of HPLC was mixed with solution of $2 \times 10^{-4}$ M NaCl in 50% (v/v %) methanol and water with a fixed flow rate at 0.15 mL/min.

For the chromatogram of $Hex_{10}GlcNAc_2$ N-glycan extracted from IgY [Figure 10b], the HPLC conditions are follows. Hypercarb column (3 μm, 2.1 × 100 mm Thermo Fisher Scientific) was used at room temperature, mobile phase gradient was changed linearly from 8% of solution A (0.1% formic acid in ultrapure water) and 92% of solution B (ACN) at $t = 0$ to 18% of solution A and 82 % of solution B at t = 35 min, and the flow rate was 0.15 mL/min. The output of HPLC was mixed with solution of $10^{-4}$ M NaCl in 80% (v/v %) methanol and water with a flow rate of 0.10 mL/min.

*Potential effects using other instrument.* When different instruments are used, one must consider that different instruments may result in different relative ion intensities in the CID spectra due to the differences in mass-dependent trapping and detection efficiencies, and the difference in the collision energy. The definition of collision energy varies between instruments and there is not always a calibration of the collision energies between different instruments. The collision energy used in the dissimilar instruments may be out of the energy range we have tested. Unfortunately, we have yet to have the opportunity to compare the performance of different instruments.

## Data availability

The data supporting the findings of this study are available within the article and its Supplementary Information.

## Code availability

No custom code or mathematical algorithm that is deemed central to the conclusions was used in this study.

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

## Acknowledgements
This work was financially supported by the thematic research project (AS-107-TP-A08) and Grand Challenge Seed Program (AS-GC-109-06) of Academia Sinica, Taiwan.

## Author contributions
C.Y.L. analyzed all chemically synthesized *N*-glycans, built the sialygalactose database, extracted and analyzed the *N*-glycans from soybeans, and wrote the manuscript and Supplementary Information. C.-C.Y. extracted and analyzed the *N*-glycans from oval-bumin. J.-L.C. extracted and analyzed the *N*-glycans from IgY. S.-T.T. analyzed the triantennary *N*-glycans. S.P. and C.-Y.W. synthesized the triantennary and tetra-antennary *N*-glycans and analyzed the corresponding NMR spectra. C.-K.N designed the experiment, conceived the LODES/MS$^n$ method, and wrote the manuscript and Supplementary Information.

## Competing interests
C.Y.L., S.-T.T., and C.-K.N. are co-inventors of a United States patent (US 10,796,788 B2) that part of the method described in the patent to determine the carbohydrate structure was used in this work. The remaining authors declare no competing interests.
