## [Peer Review File · Communications Chemistry]

Reviewers' comments:

Reviewer #1 (Remarks to the Author):

In this manuscript, the authors extend their LODES method to study complex N-glycans, which is very useful in accurate structure determination.

Reviewer #2 (Remarks to the Author):

In this work, the authors describe the application of their previously published technique, logically derived sequence (LODES) tandem mass spectrometry to the analysis of N-linked glycans. This work was validated using a series of N-glycan isomers synthesized in their laboratory and applied to the analysis of N-glycans derived from biological sources, primarily soybeans, ovalbumin, and IgY. The authors also describe the logic used for their isomer assignments based on specific fragmentation rules, and those rules are subsequently applied to both the standards and their biologically-relevant N-glycans. While the technique itself has been previously published by this research group, this appears to be their first application of LODES/MS_n to intact N-glycan isomers. The manuscript has several minor grammatical errors, but the overall scientific foundation of the work appears to be sound and is supported with sufficient detail between the figures and text. Overall, I believe that the information provided in this manuscript will benefit the glycomics community as a whole, as it provides an additional means of potentially identifying glycan isomers and allows for glycomics researchers to have an avenue of isomer identifications beyond what is currently employed in the field. In this reviewer's opinion, the work is suitable for publication in Communications Chemistry after minor revisions.

Revisions and Comments:

1. This reviewer would recommend that the authors consider restructuring the way that they introduce the dissociation mechanism rules used by LODES. The way in which Rule 2 is structured is particularly cumbersome to the reader, so breaking into individual sentences instead of one long, continuous list may be beneficial.
2. For the glycans that were synthesized within the authors' laboratory, how were the linkages of the synthetic glycans confirmed? Was NMR used or some other methodology? It may be beneficial to the reader to supply this information within the text of the manuscript, particularly given that this manuscript's primary goal is to validate the use of LODES on isomeric glycans.
3. In the source of materials section, the authors' inclusion of the soybean source on lines 346-347 seems to be randomly placed within that paragraph, given that most of the paragraph was dedicated to describing how the IgY was isolated from the hen eggs. The authors may want to consider combining all of the sources of the materials within a single paragraph, then describe the methods used for glycan release and purification within a subsequent paragraph.
4. Both the term *m/z* and the N of N-glycan should be italicized.
5. Within the description of the mass spectrometry methods, the authors state that a

“nanoelectrospray ionization instrument” (line 410) was used in their initial experiments, but they do not specify the vendor or model. This should be included to improve reproducibility of the manuscript by other researchers.

Reviewer #3 (Remarks to the Author):

This is an interesting paper describing carefully done work using tandem mass spectrometry to identify isomers of N-glycans. This is an extremely important field of high current interest and the work is thus timely.

This paper is not particularly simple to read by its very nature -- one has to stare for some time at the figures to follow the logic of their identification schemes. Nevertheless, the work seems carefully done, and I believe that their conclusions are well supported by the data.

That being said, I do have some comments about the way their approach is expressed in the paper, and I think they should have the opportunity to consider these points before the paper is finally accepted.

1) They state right at the start the three "rules" used in for structural determination. To some degree, I object to them being called "rules" in that this word seems too strong -- they are more like "propensities". This can be clearly seen in the language that they use to describe them:

- rule 1: ". . . mainly occurs . . ."
- rule 2: "much larger", "much smaller"

They then go on to use the same imprecise language in their data analysis throughout the paper.

These are not well quantified terms. How much larger is "much larger" or how much smaller is "much smaller". It seems a bit contradictory to call them "rules" and then leave the language so loose. In their programming they must put some definite limits or cut-offs to quantify these terms. They could at least be more specific on these things in the supporting information.

2) They state in Rule 3 that "Any glycosidic bond can be cleaved to produce B, C, Y, and Z ions. There is no justification given for this. Is this true in general? Does it require being able to reach certain CID energies? To make such a statement, they should justify it, and at least cite the relevant literature.

3) They should address the question of the robustness of these "rules". If different laboratories are conducting these experiments using different spectrometers, will the rules still hold if different collision energies are used. Over what range of parameters do these rules hold.

4) In figure 5a and 6b they show a chromatogram with two peaks, which they attribute to the alpha and beta anomers at the reducing end. How do they know this, and can they assign which one is which?

While I believe the authors should address these points, I think that the paper can ultimately be published.

Point-by-point response to the referees' comments

Reviewer #1 (Remarks to the Author):

Comments:

In this manuscript, the authors extend their LODES method to study complex N-glycans, which is very useful in accurate structure determination.

Reply: We thank reviewer's comment.

Reviewer #2 (Remarks to the Author):

In this work, the authors describe the application of their previously published technique, logically derived sequence (LODES) tandem mass spectrometry to the analysis of N-linked glycans. This work was validated using a series of N-glycan isomers synthesized in their laboratory and applied to the analysis of N-glycans derived from biological sources, primarily soybeans, ovalbumin, and IgY. The authors also describe the logic used for their isomer assignments based on specific fragmentation rules, and those rules are subsequently applied to both the standards and their biologically-relevant N-glycans. While the technique itself has been previously published by this research group, this appears to be their first application of LODES/MSn to intact N-glycans isomers. The manuscript has several minor grammatical errors, but the overall scientific foundation of the work appears to be sound and is supported with sufficient detail between the figures and text. Overall, I believe that the information provided in this manuscript will benefit the glycomics community as a whole, as it provides an additional means of potentially identifying glycan isomers and allows for glycomics researchers to have an avenue of isomer identifications beyond what is currently employed in the field. In this reviewer's opinion, the work is suitable for publication in Communications Chemistry after minor revisions.

Revisions and Comments:

Comment 1. This reviewer would recommend that the authors consider restructuring the way that they introduce the dissociation mechanism rules used by LODES. The way in which Rule 2 is structured is particularly cumbersome to the reader, so breaking into individual sentences instead of one long, continuous list may be beneficial.

Reply: Rule 2 was divided into several individual sentences, according to reviewer's comment.

Comment 2. For the glycans that were synthesized within the authors' laboratory, how were the linkages of the synthetic glycans confirmed? Was NMR used or some other methodology? It may be beneficial to the reader to supply this information within the text of the manuscript, particularly given that this manuscript's primary goal is to validate the use of LODS on isomeric glycans.

Reply: The NMR spectra and assignments are attached in the revised supplementary information.

Comment 3. In the source of materials section, the authors' inclusion of the soybean source on lines 346-347 seems to be randomly placed within that paragraph, given that most of the paragraph was dedicated to describing how the IgY was isolated from the hen eggs. The authors may want to consider combining all of the sources of the materials within a single paragraph, then describe the methods used for glycan release and purification within a subsequent paragraph.

Reply: In the revised manuscript, we combined all of the sources of the material in a single paragraph, described the extraction of IgY in a separate paragraph, and then described the glycan release in subsequent paragraphs, according to reviewer's comment.

Comment 4. Both the term m/z and the N of N-glycan should be italicized.

Reply: Changes were made in the revised manuscript, according to review's comment.

Comment 5. Within the description of the mass spectrometry methods, the authors state that a "nanoelectrospray ionization instrument" (line 410) was used in their initial experiments, but they do not specify the vendor or model. This should be included to improve reproducibility of the manuscript by other researchers.

Reply: "Nanospray Flex housing (Thermo Fisher Scientific)" and "The emission was monitored using a DinoLite Premier digital microscope (model AM4113TL, AnMo Electronics, Taipei, Taiwan)" were added in the paragraph of (a) Nanoelectrospray-Mass spectrometry, (B) Mass spectrometry, section of Method.

Reviewer #3 (Remarks to the Author):

This is an interesting paper describing carefully done work using tandem mass spectrometry to identify isomers of N-glycans. This is an extremely important field of high current interest and the work is thus timely.

This paper is not particularly simple to read by its very nature -- one has to stare for some time at the figures to follow the logic of their identification schemes. Nevertheless, the work seems carefully done, and I believe that their conclusions are well supported by the data.

That being said, I do have some comments about the way their approach is expressed in the paper, and I think they should have the opportunity to consider these points before the paper is finally accepted.

Comment 1) They state right at the start the three "rules" used in for structural determination. To some degree, I object to them being called "rules" in that this word seems too strong -- they are more like "propensities". This can be clearly seen in the language that they use to describe them:

- rule 1: ". . . mainly occurs . . ."

- rule 2: "much larger", "much smaller"

They then go on to use the same imprecise language in their data analysis throughout the paper.

These are not well quantified terms. How much larger is "much larger" or how much smaller is "much smaller". It seems a bit contradictory to call them "rules" and then leave the language so loose. In their programming they must put some definite limits or cut-offs to quantify these terms. They could at least be more specific on these things in the supporting information.

Reply: In the revised manuscript, the "rule" was changed to "propensity", according to the comment of reviewer. We added the following paragraph after propensity 2. "Our previous study²⁸ showed that the fragments produced from dehydration or crossed-ring dissociation at nonreducing end are typically less than 5% of the fragments produced from dehydration or crossed-ring dissociation at the reducing end, except few cases that they are up to 20%. Therefore, the aforementioned statements of 'intensity of M fragment is much larger than that of N fragment' means 'intensity of M fragment is at least 5 times larger than that of N fragment'."

Comment 2) They state in Rule 3 that "Any glycosidic bond can be cleaved to produce B, C, Y, and Z ions. There is no justification given for this. Is this true in general? Does it require being able to reach certain CID energies? To make such a statement, they should justify it, and at least cite the relevant literature.

Reply: The following paragraph and two references were added after propensity 3 in the revised manuscript. "Cleavage of glycosidic bonds produces B, C, Y, Z ions are commonly observed in sodium adducts^{33,34}, although the intensities of B and Y ions are typically larger than that of C and Z ions." We did not find significant change of the relative intensities of B, C, Y, Z ions in the CID energy we studied (normalized collision energy ranged from 25-100%).

Comment 3) They should address the question of the robustness of these "rules". If different laboratories are conducting these experiments using different spectrometers, will the rules still hold if different collision energies are used. Over what range of parameters do these rules hold.

Reply: The following paragraphs were added in the section of Method to describe the potential effects due different spectrometers and collision energies. They are at the beginning and the end of the section (B) Mass spectrometry, respectively.

LODES uses low energy CID (He as collision gas) and multistage tandem mass spectrometry for structural identification. Any mass spectrometers with these capabilities in principle can be used. We used LTQ XL linear ion trap mass spectrometer (Thermo Fisher Scientific Inc., Waltham, MA, USA) in this study simply because this instrument is readily available in our laboratory. We found that CID spectra (i.e., relative intensities of fragments) do not change significantly in the normalized collision energy ranged from 25% to 100% in the instrument we used. For most of the CID spectra shown in this study, the normalized collision energies of 30-40% were used.

When different instruments are used, one must consider that different instruments may result in different relative ion intensities in the CID spectra due to the differences in mass-dependent trapping and detection efficiencies, and the difference in the collision energy. The definition of collision energy varies between instruments and there is not always a calibration of the collision energies between different instruments. Furthermore, the collision energy used in the dissimilar instruments may be out of the energy range we have tested. Unfortunately, we have yet to have the opportunity to compare

the performance of different instruments.

Comment 4) In figure 5a and 6b they show a chromatogram with two peaks, which they attribute to the alpha and beta anomers at the reducing end. How do they know this, and can they assign which one is which?

Reply: The following paragraph and a figure [Figure 5(b)] was added in the revised manuscript to explain how we attribute two peaks in chromatogram to the alpha and beta anomers at the reducing end of the same isomer.

“We found only two peaks in the chromatogram [Figure 5(a)] after separating by porous graphitic carbon (PGC) column, the last column in multi-dimensional HPLC separation (see details in Experimental Method). PGC column is known for separating anomeric isomers, so the two peaks in Figure 5(a) might be the alpha and beta anomers of the same Hex₈GlcNAc₂ N-glycan isomer. To clarify this, the eluents separated from the PGC column were collected in fraction every 30 seconds so that compound corresponding to each peak in Figure 5(a) was collected into different test tubes. The same collection was repeated for 10 times which the eluents of the same retention time were combined in the same test tube. Then, these fractionated eluents were stored at room temperature for more than 6 hours before they were concentrated and reinjected into the same PGC column (with the same separation gradient) individually. If the two peaks in Figure 5(a) belong to one isomer and they are only different by the anomericity at the reducing end, the reinjection of the fractionated eluents into the same PGC column would show two peaks in chromatogram, and the relative intensities and the retention times of these two peaks must be the same as that in Figure 5(a). This is because the anomers undergo mutarotation, i.e., change from α (or β) configuration to β (or α) configuration, in solution and this mutarotation typically takes only about 30 min-2 hours at room temperature. Although each fraction contains only one anomer when it was collected right after the PGC separation, the anomer undergoes mutarotation during the 6-hour storage time. As a result, there are two anomers in each collected fraction and these two anomers reach equilibrium. The PGC chromatograms of these fractionated eluents, as illustrated in Figure 5(b), indeed shows two peaks at the same retention time with the same relative intensities as that in Figure 5(a). Tube 10 (fraction collected at 21.5- 22 minutes) and tube 14 (fraction collected at 23.5- 24 minutes) have the largest intensity among all the collected fractions, as they correspond to the fractions collected at the retention times of two peaks in the first PGC chromatogram [Figure 5(a)], respectively. Consequently, we can conclude that the two peaks

in the first PGC chromatogram [Figure 5(a)] belong to one isomer and they are only different by the anomericity at the reducing end. The structure was identified as 8A_{1,2,3}B_{1,3}. Details of the CID spectra for the structural determination are presented in the Supplementary Figure S4.”

Detailed description and chromatograms were added in the revised manuscript for soybean N-glycans. Similar procedure was performed to ensure two peaks in Figure 6(b), the N-glycan of IgY, belong to one isomer. Details were described in the supplementary. We are not able to assign the anomericity at the reducing end because the dissociation property of HexNAc i.e., α and β configurations of HexNAc lead to the same CID spectrum which has been described in our previous published paper (reference 31).

REVIEWERS' COMMENTS:

Reviewer #2 (Remarks to the Author):

All of my previous comments on this manuscript have been appropriately addressed, and I am satisfied with the changes. In this reviewer's opinion, this paper is now suitable for publication in Communications in Chemistry.

Reviewer #3 (Remarks to the Author):

I am completely satisfied with the changes made by the authors in this revised manuscript, and I now fully support publication of this report.